# Cyclosporine A Delivery Platform for Veterinary Ophthalmology—A New Concept for Advanced Ophthalmology

**DOI:** 10.3390/biom12101525

**Published:** 2022-10-20

**Authors:** Martyna Padjasek, Badr Qasem, Anna Cisło-Pakuluk, Krzysztof Marycz

**Affiliations:** Department of Experimental Biology, The Faculty of Biology and Animal Science, The University of Environmental and Life Sciences, 50-375 Wroclaw, Poland

**Keywords:** cyclosporine A, keratoconjunctivitis sicca, chronic superficial keratitis, immune-mediated keratitis, equine recurrent uveitis, delivery devices

## Abstract

Cyclosporine A (CsA) is a selective and reversible immunosuppressant agent that is widely used as a medication for a wide spectrum of diseases in humans such as graft versus host disease, non-infectious uveitis, rheumatoid arthritis, psoriasis, and atopic dermatitis. Furthermore, the CsA is used to treat keratoconjunctivitis sicca, chronic superficial keratitis, immune-mediated keratitis and equine recurrent uveitis in animals. The selective activity of Cyclosporine A (CsA) was demonstrated to be an immunomodulation characteristic of T-lymphocyte proliferation and inhibits cytokine gene expression. Moreover, the lipophilic characteristics with poor bioavailability and low solubility in water, besides the side effects, force the need to develop new formulations and devices that will provide adequate penetration into the anterior and posterior segments of the eye. This review aims to summarize the effectiveness and safety of cyclosporine A delivery platforms in veterinary ophthalmology.

## 1. Introduction

Cyclosporine A (CsA) is one of the most important transplantation drugs that was discovered and isolated by Jean Borel and co-workers in 1970, from the fungus *Tolypocladium inflatum* [1,2]. In 1976, their results demonstrated that cyclosporine has immunosuppressive characteristics [3], which were crucial in transplantology and immunopharmacology.

Nowadays, it is widely used as human and veterinary medicine in transplantation procedures, and some immune-mediated inflammatory diseases (IMIDs) [4]. Nevertheless, it has several side effects. For instance, a recent study demonstrated that up to 50% of patients have CsA-associated neurotoxicity in both intravenous and oral administrations which makes the CsA mechanism of action remain ambiguous [5].

This review aims to summarize the effectiveness and safety of CsA delivery platforms in veterinary ophthalmology. However, it is important to highlight numerous studies and great progress in human ophthalmology such as the intracameral drug-delivery system for high-risk penetrating keratoplasty [6,7], modified intraocular lens to inhibit post-cataract surgery uveitis and preventing posterior capsular opacification [8,9,10,11], silica-thermogel nanohybrids sustainably releasing drugs after subconjunctival injection [12,13], or a glaucoma drainage device containing CsA and poly(lactic-co-glycolic acid) (PLGA) to prevent postoperative fibrosis [13].

## 2. Cyclosporine A: The Physicochemical Properties

Cyclosporine A is a natural cyclic hydrophobic peptide with eleven amino acid residues (cyclo[MeBmt^1^-Abu^2^-Sar^3^-MeLeu^4^-Val^5^-MeLeu^6^-Ala^7^-d-Ala^8^-MeLeu^9^-MeLeu^10^-MeVal^11^], seven peptide bonds (N-methylated), four intra-molecular hydrogen bonds responsible for the cyclic structure, and a molecular weight of 1202.6 g/mol (Figure 1) [14].

The CsA is highly soluble in organic solvents such as methanol, ethanol, acetone, ether and chloroform, yet with a different degree of solubility in each one of those solvents, according to Gonzalez et al. and Czogalla et al. [15,16]. For instance, the CsA has the lowest solubility in water with 0.04 mg/g. However, the highest solubility occurred in chloroform, acetonitrile, dimethyl sulfoxide, methanol, ethyl acetate, isopropyl alcohol, ethanol, polyethylene glycol acetate, isopropyl alcohol, ethanol, polyethylene glycol, propylene glycol, N,N-dimethylacetamide, glycofurol 75, N-methylpyrrolidone, sesame oil, labrafil, labrafac, oleic acid, Tween-20, and Solutol HS with >100 mg/g [15,16].

In addition, there are some parameters that require further consideration. For instance, the temperature and the pH. The temperature-dependent parameter works in an inversely proportional way; thus, the CsA solubility in water at temperatures ranging between 5 °C to 37 °C showed the highest solubility at 5 °C (101.5 µg/mL) and the lowest at 37 °C (7.3 µg/mL). Therefore, the behavior mechanism associated with D-Ala amino acid residual position number 8 (Figure 1) because of its hydration water is lost at high temperatures. Moreover, the selected pH values that represent the pH range of the stomach and the small intestine showed no significant effect on the CsA solubility at pH 1.2 and 6.6 [17]. Poor membrane permeation after topical application, especially with oral administration, related to low water solubility linked with high lipophilic characteristics (log *P* = 2.92 at pH 7.4), and significant rigidity of the cyclic structure of the CsA results in a limited absorption of the peptide across the gastrointestinal membrane. For that reason, Cyclosporine A (CsA) was classified as a Class IV under the biopharmaceutics classification system [18,19]. Therefore, the proper selection of drug administration is essential for successful therapy.
Figure 1(**a**) The chemical structure of Cyclosporine A (C_62_H_111_N_11_O_12_); (**b**) The crystal structure of CsA. Blue—nitrogen; red—oxygen; gray—carbon; proton atoms—white [20,21].
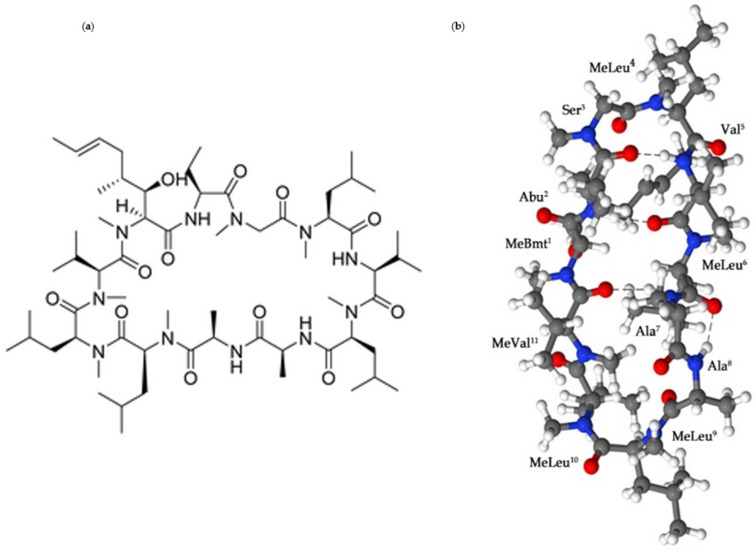


## 3. The Mechanisms of Action

Cyclosporine A (CsA) affects the proliferation of helper subset T lymphocytes and cytokine production. The mechanism is linked to two major CsA pathways on the calcineurin/NAFT pathway, as well as JNK and p38 signaling pathways [22,23,24].

Cyclosporine A blocks the T cells’ infiltration and subsequently the expression of the inflammatory cytokines such as IL-2 and IL-4 via the calcineurin/NAFT pathway by cyclosporine-cyclophilins interaction in the cytoplasm of T-cells, which causes an increase in Ca^2+^ in the cell. A high concentration of Ca^2+^ combined with an inducing T cell receptor (TCR) activates calmodulin and binds to protein serine/threonine phosphatases known as calcineurin (calmodulin-dependent protein phosphatase).

The calcineurin has catalytic (CnA) and regulatory (CnB) subunits. However, the calcineurin catalytic (CnA) is the dominant in T-cells.

The calmodulin- calcineurin A interaction causes the inhibitor domain active site in the CnA to be released and inhibits the phosphatase activity. Therefore, the combination mechanism of the cyclosporine-cyclophilin binding to calcineurin A dephosphorylates the nuclear factor of activated T cell (NFAT) family members (NFAT1, NFAT2, and NFAT4), and prevents the translocation of NAFT family members into the nucleus and transcription of lymphokines genes [23,25] (Figure 2).

The JNK, p38 and ERK signaling pathways are subgroups of the mitogen-activated protein kinase (MAPK) superfamily. However, JNK and p38 signaling pathways showed a greater selectivity of effect to cyclosporine A (CsA) compared to the ERK pathway. Both pathways are synergistically activated during stress responses, such as inflammation and apoptosis, as well as when T cells are stimulated by TCR and CD28 costimulatory receptors. It has been demonstrated that any mutation that blocks JNK and p38 signaling pathways revokes the NF-AT cis-element transcription activation which has binding sites for NFAT family members and Activator protein 1 (AP-1), and both are involved in IL-2 expression [26]. Moreover, the inhibition of both JNK and p38 signaling pathways is related to the upstream level of MAPKK-K activation for example, the MEKK1/MAPKK-Ks participates in the JNK and p38 signaling pathways through MKK7 and MKK6 [18]. Other potential indirect actions of CsA suggested that the JNK and p38 signaling pathways inhibition could be related to over-expression of Vav1/Vav2/Dbl and GEF for Rac1 or Cdc42. Moreover, the HPK1 (Rac1-independent) contributes to JNK activation in an indirect manner. However, the CsA mechanism of action on JNK and p38 signaling pathways remains obscure [27,28] (Figure 2).

## 4. Drug Delivery of Cyclosporine

Systemic administration of cyclosporine is commonly used in transplantation procedures but also showed effectiveness in topical treatment, especially during the inflammatory process of the eye [29]. In ophthalmic diseases, direct ocular administration is preferable because of the systemic administration-related side effects such as nephrotoxicity, digestive tract disorders and hypertension [30,31,32]. There are two topical formulations of CsA registered in human medicine for the treatment of keratoconjunctivitis sicca (KCS): Restasis^®^ in the USA and Ikervis^®^ in Europe [29].

In veterinary ophthalmology, Optimmune ^®^ ointment containing 0.2% cyclosporine is registered for analogous purposes and additionally for chronic superficial keratitis (CSK) [33]. However, there are many examples of immune-mediated ocular disorders in which cyclosporine is beneficial [34]: graft versus host disease (GVHD) [35], recurrent anterior uveitis [36], vernal keratoconjunctivitis [37] in humans, and equine recurrent uveitis [38] and immune-mediated keratitis [39] in horses.

Because of the physiochemical properties of the CsA, new formulations and devices are investigated [22,29]. To ensure high penetration capability, long-term effects, and constant drug delivery or to minimize problems with administering eye drops by animal owners, CsA-incorporated implants were developed.

## 5. CsA-Implants in Veterinary Ophthalmology

Pearson et al. formulated one of the first CsA delivery devices [40], based on a sustained-release ganciclovir intravitreal implant [41].

The device containing 5 mg of CsA was implanted intravitreally in eighteen New Zealand albino rabbits and an analogous implant containing 6 mg of CsA was used in three cynomolgus monkeys. The study aimed to determine the toxicity of an intravitreal device that provides long-term delivery of CsA. However, the results showed no evidence of toxicity in the cynomolgus monkeys, but in the rabbits lens opacification in the vicinity of the implant was observed as well as a decrease of the b-wave amplitude in the ERG [40].

In addition, a similar study was established by Enyedi et al. [42], who investigated the intraocular device containing a combination of dexamethasone (2 mg) and CsA (100 ug) in New Zealand albino rabbits. The 2.5 mm diameter drug pellet of the implant was coated with polyvinyl alcohol (PVA) and ethylene vinyl acetate (EVA) [40,41]. The highest levels of CsA were detected in the lens compared to low levels in the sclera, cornea, iris and aqueous [42]. Moreover, Jaffe et al. have determined the effectiveness of the intravitreal CsA-sustained delivery device, that proposed by Pearson et al., in the treatment of experimental uveitis in rabbits. The inflammation in the treated eye was considerably less than in the control eye. The therapeutic level of CsA in the vitreous was detected 6 months after implantation [43].

### 5.1. Keratoconjunctivitis Sicca (KCS)

Keratoconjunctivitis sicca is an inflammatory disease that affects the gland of the third eyelid and lacrimal glands, causing a decrease of tear film, quantitative or qualitative disorder, that could be the result of a congenital, metabolic, drug-induced, neurogenic or immune-mediated defect [44]. The most common signs of KCS are mucopurulent ocular discharge, conjunctivitis and keratitis which can lead to corneal ulcers; some patients develop blepharitis and, in chronic cases, corneal pigmentation and scarring occur [45,46]. An indispensable element of ophthalmic examination is the Schirmer test (STT-1) which in normal dogs shows tear production around 15 to 25 mm/min; STT-1 in the course of KCS ranges from 9–14 mm/min as mild, >4 to 8 mm/min as moderate and <4 mm/min as a severe stage [47].

The histopathology and serologic results suggested that most of the cases may be immune-mediated (more than 30% of dogs with KCS); the confirmation of which is a positive reaction to immunosuppressive drugs [44,45,48,49].

Kim et al. formulated three similar silicone-based matrix CsA implants (with 20–30% *wt*/*wt*), which were used in several subsequent studies. For instance, the study aimed to provide drug delivery devices that were effective in treating lacrimal gland GVHD (graft versus host disease after transplantation of allogeneic stem cells) and assessed the rate of CsA release (in vitro), implant toxicity, pharmacokinetics and pharmacodynamics in normal rabbits, dogs and dogs with KCS. The results after six months revealed the safety of the delivery device; no ocular toxicity and abnormalities in blood examination were observed. The implant provided therapeutic levels of CsA in the lacrimal gland, conjunctiva and cornea; dogs (with clinical signs of KCS and Schirmer test below 5 mm/min) after implantation did not need further local treatment and the Schirmer test results were above 10 mm/min during the study period [50].

Acton et al. reported a case of keratoconjunctivitis sicca in a red wolf (*Canis rufus*) [51]. After a positive response to topical 2% cyclosporine (initial Schirmer test result at 0 mm/min and after two weeks of combination therapy with triple antibiotic with dexamethasone, the tear production levels were 15 mm/min in the left eye and 16 mm/min in the right eye) they performed implantation of episcleral sustained-release CsA devices (10% matrix CsA-silicone). After two weeks of implantation the tear production remained at the physiological level above 13 mm/min twelve months after surgery [51].

Penetrating keratoplasty (PKP) is a common allograft performed in humans. However, there is one major drawback of this procedure—its high rejection rate (65%). To overcome this obstacle Lee et al. have proposed the use of episcleral CsA implant [52] described previously: CsA powder mixed with silicone; *wt*/*wt* 30% [49]. Two implants with different total release were used to determine short (implant B—7.7 mg CsA per implant) and long-term (implant A—12 mg CsA per implant) pharmacokinetics in rabbits and dogs, therefore, the cumulative release observed over the 400-days was approximately 3.8 mg (implant A) and 2.3 mg (implant B). This study showed effective penetration into the cornea and no signs of ocular toxicity. Moreover, CsA concentrations in the cornea were approximately 0.1 µg/mg three hours after implantation and ensured the suppression of T-cell and vascular endothelial cells for over a year. Pharmacokinetics evaluation of CsA in the rabbit model was detected in buccal lymph nodes at 1 h, which suggests that lymphatic vessels in conjunctiva support the rapid dissolution of the drug to the cornea and surrounding tissue [52].

Numerous studies have shown the effectiveness and safety of a cyclosporin episcleral implant with a silicone matrix, an implant of 1.9 cm × 2 mm × 1 mm, containing 12 mg of CsA and ensuring its release at an average level of 17 µg/day for at least 6 months is particularly useful in anterior segment disease [50,53].

Choi et al. proposed hydrogel contact lenses (CLs) loaded with CsA and determined its efficiency in the rabbit model of dry eye [54]. Previous studies using drug-soaked lenses showed low efficiency in sustained release of the drugs [55,56,57,58,59]. Therefore, they used a supercritical fluid (SCF) technique to modify and control the degree and the rate of releasing CsA. The in vivo study showed that adequate concentration of CsA was maintained for over 48 h in the cornea, conjunctiva, and crystalline lens. In comparison with control groups, the CsA-CL group exhibited higher density of the goblet cell, tear volume, lower staining score, and reduction of the inflammatory process through immunomodulatory effects.

Several clinical trials presented new methods for extended-release drug delivery. For Sight Vision, owned by Allergan, proposed a peri-conjunctival ring currently used for delivery bimatoprost in glaucoma patients [60]. Work is well under way to deliver a CsA ring based on the same technology. In cooperation with NC State University, they conducted clinical trials on dogs with KCS, in which a conjunctival ring releasing CsA was well tolerated and as long as it rested on the conjunctiva under the upper and lower eyelids, the results were satisfying. The therapeutic effect lasted about a month with 75% retention in the eye (unpublished data, ESVO Webinar, 21 February 2021).

Ocular Therapeutix™ is working on a group of drug-eluting intracanalicular drug inserts. A study funded by Ocular Therapeutix™ and conducted by Vanslette et al. evaluated pharmacokinetics of Cyclosporine Intracanalicular Insert (OTX-CSI) in Beagle dogs with surgically induced Dry Eye. Intracanalicular devices combines two treatments of dry eye disease: sustained release delivery of cyclosporine and punctal occlusion which aids tear conservation. OTX-SCI contains 0.36 mg CsA in fully biodegradable polyethylene glycol hydrogel. It was designed to provide effective therapy for 12 weeks. The study showed successfully released CsA and its higher concentration in tear fluid in dogs with dry eye was probably due to less dilution on the ocular surface. OTX-CSI was well tolerated and assured immunomodulatory levels in tear fluid [61]. Although OTX-CSI is a promising device for the treatment of immunological diseases in human and veterinary ophthalmology, Ocular Therapeutix™ published results of a Phase 2 clinical trial in which OTX-SCI did not meet the primary endpoint of increased tear production at 12 weeks; therefore, more research is required.

### 5.2. Chronic Superficial Keratitis (CSK)

Chronic superficial keratitis, also known as Pannus, is another immune-mediated disease that affects dogs, with chronic corneal lesions characteristic, mostly in the lateral quadrant: vascularization, progressive pigmentation, and sometimes white opacity [61,62,63]. The etiology is still not fully understood but CD4+ T lymphocyte infiltration from cornea stroma suggests an immunological background [64,65,66,67]. The German shepherd dog is a predisposed breed, but it can also occur in the Australian shepherd, collie, border collie, golden retriever, Akita, vizsla, and others [62,68]. Several studies have found that excessive ultraviolet exposure increases the risk of CSK [62,69,70].

Topical immunomodulators such as steroids and calcineurin inhibitors are the standard therapy [71]. Dogs that are responsive to topical CsA, and for whom therapy must be continuous, might be good candidates for an episcleral cyclosporine implant proposed for treating keratoconjunctivitis sicca. However, further studies are required [50,52].

### 5.3. Immune-Mediated Keratitis (IMMK)

Immune-mediated keratitis is nonulcerative, primary keratitis (NUK) that occurs in horses [72]. Although the etiopathology has not been thoroughly investigated, the absence of microorganisms and significant improvement after implementation of immunosuppressive therapy suggest an immunological background. The common symptoms of IMMK are nonulcerative corneal opacity, corneal edema and neovascularization, cellular infiltration, and no features of uveitis. Horses with IMMK experience no or mild discomfort [39]. IMMK is classified into four types based on its location in the cornea: epithelial, superficial stromal, middle stromal, and endothelial, with the superficial stroma being the most frequent site of occurrence [72]. CsA topical application is most effective in the case of epithelial and superficial IMMK, but efficiency decreases along with the posterior layers of the cornea [39]. Thus, another route of CsA distribution to all layers of the cornea is being investigated.

Gilger et al. proposed the use of a silicone matrix CsA episcleral implant in nineteen horses with different types of IMMK [73]. More than two devices, described previously [50,52], were implanted per eye in the dorso temporal episcleral space.

The study demonstrated good tolerance of the implants with no significant deviations between the number of devices implanted. Superficial and endothelial immune-mediated keratitis were considered controlled in all treated eyes, although in three cases of endothelial IMMK topical bromfenac was also administered. The worst response was observed in the case of midstromal IMMK. Implants were unable to control inflammation. In vivo, the therapeutic effect of CsA in the case of superficial IMMK was determined for 12–18 months [73].

### 5.4. Equine Recurrent Uveitis (ERU) 

Equine recurrent uveitis (moon blindness, periodic ophthalmia, iridocyclitis) is a condition in which immune-mediated active episodes of panuveitis reoccur every few weeks to months [38]. ERU is still one of the most common causes of horse blindness. Around 30% of horses who presented for examination due to ERU symptoms were unilaterally or bilaterally blind [74,75]. Horses experience spontaneous relapses similar to humans [76,77,78]. Initial causes of recurrent uveitis are not always known, but genetic predisposition or microbes such as *Leptospira* sp. might be involved [78,79]. According to recent research, the retinal expression of neuraminidase 1 (NEU1) plays an important role in ERU. Furthermore, horses with recurrent uveitis had higher levels of NEU1 in Müller glial cells in the retina. Therefore, NEU1 might be a new marker of activated Müller glial cells in uveitis [80].

Clinical signs associated with ERU can include anterior segment: blepharospasm, increased lacrimation, photophobia, miosis, edema and vascularization, aqueous flare, cellular infiltration, hypopyon and hyphema, low IOP; posterior segment: vitreous, chorioretinitis, and retinal degeneration [78,81].

Gilger et al. used an intravitreal cyclosporine delivery device, previously described by Pearson et al., 1996 [40], Enyedi et al., 1996 [42], and Jaffe et al., 1998 [43], in horses with experimental uveitis [82]. The study found that the CsA intravitreal implant reduced the severity and duration of symptoms (but the inflammatory suppression was incomplete), cellular infiltrate was less intense compared to the control eye (PVA/EVA devices without CsA), and the CsA-delivery device was well tolerated. Moreover, the concentration of cyclosporine in the vitreous humor was below therapeutic levels. Nevertheless, tissue levels were not measured [82]. Additionally, the long-term study shows that intravitreal sustained-release CsA delivery devices are safe for at least 12 months [83].

A similar implant was evaluated in horses with ERU that occurred naturally. Devices releasing 4 µg of CsA per day (in a previous study 2 µq/d) were implanted in the eyes of sixteen horses with unilateral uveitis and history of disease recurrence. Follow-up was performed between 6 and 24 months after implantation. After surgery, less than 20% of horses developed uveitis, but as reported by owners, the symptoms were less severe and responded better to anti-inflammatory medication. Complications were noted in four patients, including vision loss due to cataracts or complete retinal detachment as well as glaucoma [84].

A different study by Gilger et al. evaluated episcleral and deep scleral bioerodible cyclosporine implants [84]. Intravitreal delivery devices showed some good results but also revealed complications after implantation, such as cataracts caused by lens injury, endophthalmitis or increased risk of retinal detachment [40,43,82,83,84,85]. Thus, the use of implants that do not require entry into the eye has been proposed [50,86,87,88,89,90].

Gilger et al. conducted an in vitro study of transscleral diffusion of CsA from a biodegradable matrix-reservoir CsA implant, formulated by Robinson et al. from the National Eye Institute, that suggested the release duration of CsA around 38 months and poor penetration through the sclera. This study aimed to determine the pharmacokinetics and safety of episcleral as well as deep scleral lamellar CsA devices in horses. However, episcleral implantation of the device did not reduce the frequency of relapses due to limited penetration through the sclera. Moreover, the CsA concentration in retina-choroid and vitreous was below the required minimum to treat inflammation [40]. In addition, the deep sclera CsA device was well tolerated, and no toxicity was observed. Therapeutic drug concentration was observed in vitreous and sclera, choroid-retina and optic nerve tissue, although there was no detection of CsA in the aqueous humor, cornea, and samples of peripheral blood. Follow-ups were performed on average after 14 months and a reduction in flare-ups was noted.

Blindness occurred in 15% of the eyes as a result of glaucoma, uncontrolled uveitis, cataract, fungal keratitis, and retinal detachment. At the end of the study period, 68/80 of the eyes had vision after surgery [91]. A long-term study on 133 horses (151 eyes) confirmed the promising results from the previous survey but also noted complications such as glaucoma, persistent uveitis, cataracts, and retinal detachment [74,92,93,94].

## 6. Conclusions

Cyclosporine is a calcineurin inhibitor with immunomodulatory and immunosuppressive properties. Lipophilic, high molecular weight, poor solubility in water, and numerous side effects require an urgent need to develop new formulations and devices to deliver this therapeutic agent. The implants described so far were well tolerated and provided therapeutic levels in the target tissues. In cases of KCS, they ensured increased tear production, while controlling inflammation in other diseases. Positive results in animals promise better treatment prospects for not only graft versus host disease in humans, but also keratoconjunctivitis sicca and recurrent uveitis. However, longer and more thorough examinations may be necessary to obtain the most effective and sustainable delivery devices, ensuring a long-lasting and constant supply of the drug and high penetration, which are the critical factors for the therapy’s effectiveness and the elimination of the problem of administering drugs to uncooperative animals. The implants proposed so far require anesthetized patients and an invasive surgical procedure. To reach posterior segments of the eye, deep sclera devices are needed which come with the risk of side effects such as cataracts and high intraocular pressure. Accordingly, the future requires biodegradable implants with a long duration of action and a minimally invasive implantation procedure. Furthermore, episcleral injection-based hydrogel carriers seem to be a promising solution; therefore, there is an urgent need for further research on the form of cyclosporine administration and cyclosporine carriers.

## Figures and Tables

**Figure 2 biomolecules-12-01525-f002:**
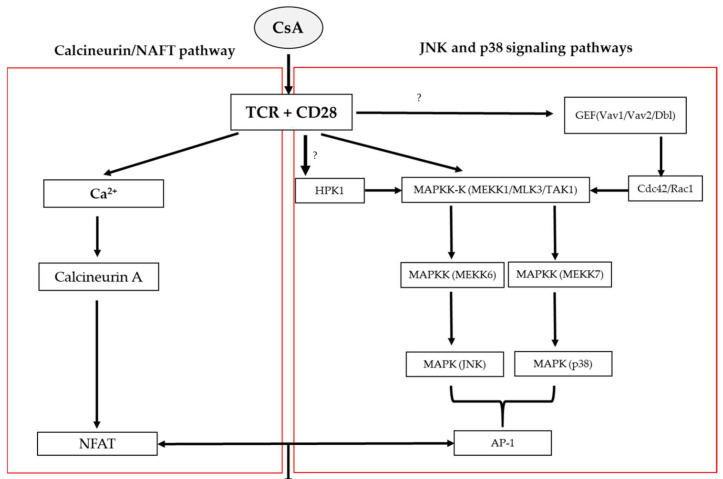
The mechanism of CsA action on the calcineurin/NAFT pathway and JNK and p38 signaling pathways in T cells. CsA, cyclosporine-A; T cell receptor, TCR; Guanine nucleotide exchange factor, GEF; Vav Guanine Nucleotide Exchange Factor 1, 2, Vav1, Vav2; diffuse B-cell lymphoma, Dbl; Hematopoietic progenitor kinase, HPK1; Mitogen-Activated Protein kinase kinase kinase, MAPKK-K also known as MEKK1/MLK3/TAK1; Mitogen-Activated Protein kinase kinase, MAPKK also known as MEKK6/MEKK7; The c-Jun N-terminal kinase pathway, JNK; Activator protein 1, AP-1; Cell Division Cycle 42, Cdc42; Ras-related C3 botulinum toxin substrate 1, Rac1; Nuclear factor of activated T-cells, NFAT.

## Data Availability

Not applicable.

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
