# Peer review of "Cyclosporine A Delivery Platform for Veterinary Ophthalmology—A New Concept for Advanced Ophthalmology"

_biomolecules, 2022, doi:10.3390/biom12101525_

Round 1
Reviewer 1 Report
I have read the manuscript “Cyclosporine A delivery platform for veterinary ophthalmology – a new concept for advanced ophtamology” by M. Padjasek et al. with great attention. The co-authors carefully collected and presented study data on cyclosporine A from the last 45 years. They also presented some general facts about this peptide (physicochemical properties, mechanism of action) and its application in ophthalmology. In the conclusion, the co-authors mention the positive aspects of the current application techniques and the drug itself, but also points out the shortcomings that modern science still has to catch up, which makes this paper very motivating for scientists.
Therefore, after a minor revision, I would like to recommend it for publication in Biomolecules. The points were listed as follows: i) Add the chemical formula of the compound along with the molecular weight for clarity (C62H111N11O12); ii) I recommend drawing Figure 1.b in the colours characteristic for the atoms of the elements presented, as this is much clearer for the reader; iii) Raws 126 and 127: “Due to side effects after systemic administration, such as nephrotoxicity, digestive tract disorders, and hypertension.”, please write this sentence correctly, put it in context; iv) Title Summery replace with Conclusions; v) In the sentence in Conclusions “Hydrophobic and lipophilic character, high molecular weight, poor solubility in water, and numerous side effects... “, is it necessary to write hydrophobic and poor solubility in water in the same sentence?; vi) There are some errors in this manuscript, such as double line spacing, comma instead of period (raw 61), delete reference 41 at the beginning of the section. Please check the whole Manuscript again.
Reviewer 2 Report
This manuscript represents a modest review of the use of cyclosporine A in veterinary medicine. The focus is primarily on its use to treat ophthalmic conditions. Some background of the molecular actions of cyclosporine are provided. The main thrust of the review is to describe development and use of sustained delivery devices for use in the eye. The review of the work in the field appears to be quite complete and up to date and should be of interest to researchers in the field as well as veterinary practitioners.
My primary criticism of the work is that the manuscript requires a thorough editing for English usage and style. In some places, the clarity of the writing can be improved. I have attached a commented copy of the manuscript with my suggestions for revision and improvement; however, I strongly recommend that the paper be edited by someone with proficiency in English writing before being resubmitted to this Journal.

Round 2
Reviewer 2 Report
Critique:
The authors have provided a thoroughly re-written version of the manuscript that greatly improves the readability and clarity. The review itself is quite thorough and veterinary ophthalmologists and researchers in the field should find it quite informative. There are no major remaining issues with the overall writing, and only some minor items remain to be corrected, as enumerated below.
Page 2, line 86: By "Dal", do you mean an abbreviation for D-Alanine? I suggest you use D-Ala as the abbreviation for greater clarity. This also correlates better with the labels on the molecular structure shown in Figure 1.
Figure 1: Will this figure be printed in color? If not, then the monochrome version in the original manuscript may be a better option for this figure.
Page 6, line 784: Replace “assured” with “exhibited”. (Or similar word)
Page 6, line 788: Change “...Allergan proposing” to “...Allergan, proposed”.
Page 6, line 791: Insert “a” before “conjunctival”
Page 6, line 796: Change “Study...” to “A study…”
Page 6, line 799: Suggest inserting “sustained” before “release”
Page 6, line 801: Change “Study…” to “The study…”
Page 6, line 803: Insert “was” before “probably”
Page 6, line 807: Change “didn’t” to “did not”
Page 6, line 807: Change “...12 weeks, so it…” to “12 weeks; therefore, it…”
Page 8, line 1182: Delete the apostrophe after the word, “owners”
Page 8, line 1196: Change “Episcleral” to “episcleral”
Reference list:
Ref #20 is not in the proper form. This reference is a book and should be cited in the proper MDPI format:
[List of chapter authors] In Peptides Synthesis, Structures, and Applications, Gutte B, Ed., Academic Press, San Diego, CA, USA, 1995, Chap. 5, pp 193-245. [Personal note: The MDPI citation format does not specifically indicate the DOI for books should be shown; however, I think including the DOI is useful for tracking down the specific reference and I suggest including it here. The copy editor may or may not agree.]
